# An On-Demand pH-Sensitive Nanocluster for Cancer Treatment by Combining Photothermal Therapy and Chemotherapy

**DOI:** 10.3390/pharmaceutics12090839

**Published:** 2020-09-02

**Authors:** Taehoon Sim, Chaemin Lim, Ngoc Ha Hoang, Yuseon Shin, Jae Chang Kim, June Yong Park, Jaewon Her, Eun Seong Lee, Yu Seok Youn, Kyung Taek Oh

**Affiliations:** 1Department of Pharmaceutical Sciences, College of Pharmacy, Chung-Ang University, 221 Heukseok dong, Dongjak-gu, Seoul 06974, Korea; simtaela@outlook.kr (T.S.); chaemin201@cau.ac.kr (C.L.); hoangngocha@cau.ac.kr (N.H.H.); sus9417@cau.ac.kr (Y.S.); wofl2018@cau.ac.kr (J.C.K.); oong93@cau.ac.kr (J.Y.P.); hjwon93@cau.ac.kr (J.H.); 2Department of Biotechnology, The Catholic University of Korea, 43 Jibong-ro, Bucheon-si, Gyeonggi-do 14662, Korea; eslee@catholic.ac.kr; 3School of Pharmacy, Sungkyunkwan University, 2066 Seobu-ro, Jangan-gu, Suwon, Gyeonggi-do 16419, Korea; ysyoun@skku.edu

**Keywords:** gold nanorod, photothermal therapy, chemotherapy, cancer, pH sensitive polymer, controlled release, nanocluster

## Abstract

Combination therapy is considered to be a promising strategy for improving the therapeutic efficiency of cancer treatment. In this study, an on-demand pH-sensitive nanocluster (NC) system was prepared by the encapsulation of gold nanorods (AuNR) and doxorubicin (DOX) by a pH-sensitive polymer, poly(aspartic acid-graft-imidazole)-PEG, to enhance the therapeutic effect of chemotherapy and photothermal therapy. At pH 6.5, the NC systems formed aggregated structures and released higher drug amounts while sustaining a stable nano-assembly, structured with less systemic toxicity at pH 7.4. The NC could also increase antitumor efficacy as a result of improved accumulation and release of DOX from the NC system at pH_ex_ and pH_en_ with locally applied near-infrared light. Therefore, an NC system would be a potent strategy for on-demand combination treatment to target tumors with less systemic toxicity and an improved therapeutic effect.

## 1. Introduction

Combination therapy has been considered to be a promising strategy for improving the therapeutic efficiency of cancer treatment [1,2,3,4]. A combination of various drugs and therapeutic modalities may synergistically inhibit cancer growth [5,6,7,8,9,10,11]. Photothermal therapy (PTT) causes cytotoxicity against malignant and diseased cells by combining a photosensitizer and a light source. Near-infrared (NIR) laser-mediated photothermal therapy has attracted interest for melanoma therapy due to its deep tissue penetration and slight absorbance [12,13,14,15,16]. Photothermal therapy also has potential use for other tumors, since additional equipment such as the optic fiber may irradiate other parts of the body [17,18,19]. Among the various photothermal agents, gold nanorods (AuNRs) have been comprehensively researched due to their excellent biocompatibility and tunable surface plasmon resonance (SPR) properties for converting NIR light into local heat [20,21,22]. A combination of AuNR-based PTT and synchronized chemotherapy using a stimuli-responsive drug delivery system for controlled drug release has been anticipated to have superior anti-cancer efficiency.

Polymeric biomaterials have been utilized as advanced drug carriers [23,24,25,26,27], because their properties can be controlled by small changes in environmental conditions [24,28,29,30,31,32,33,34,35], to improve therapeutic efficacy and minimize undesirable side effects [36,37,38,39,40,41]. Among the various polymeric biomaterials, pH-sensitive polymers have been developed as promising drug carriers for pH-dependent drug release in distinctive tumor microenvironments, such as tumor extracellular pH (pH_ex_, ranging from 6.5 to 7.2) and endosomal pH (pH_en_ ≤ 6.5) [32,42,43,44,45,46]. In particular, nanocarriers based on pH-sensitive biomaterials have shown outstanding in vitro and in vivo tumor inhibition by higher drug release. In our previous report, poly(aspartic acid)-PEG grafted by imidazole moieties (PAIM-PEG) was shown to be a pH-sensitive nanocarrier for cancer treatments (pK_a_: 6.5) [43,47,48,49]. Poly(aspartic acid) grafted by a pH-sensitive imidazole moiety, PAIM-PEG, formed rigid nanosized particles loaded with anticancer agents at pH 7.4. Decreasing the pH to pH_ex_ or pH_en_ led to particle breakdown, and the resulting drug released to the tumor site effectively inhibited tumor growth [43,47,48,49].

In this study, AuNRs and doxorubicin (DOX) were encapsulated by a pH-sensitive polymer, PAIM-PEG, to enhance the therapeutic effect of chemotherapy and photothermal therapy through tumor-targeted drug delivery and controlled drug release. Considering the previous report on the pH sensitivity of PAIM-PEG (pK_a_: 6.5), which is theoretically more sensitive than the hydrazone bond (pK_a_: ~5.5; [50,51]), combination therapy using a nanocluster would be on-demand for cancer treatment. Preparation of a nanocluster (NC) system, physicochemical characterization, and in vitro and in vivo therapeutic assessments were conducted to confirm the potential of the pH-sensitive nanocluster for the feasibility of tumor reversal.

## 2. Materials and Methods

### 2.1. Materials

AuNRs (gold nanorods functionalized with amine groups, 10 nm (diameter), 808 nm (absorption max)) were purchased from Creative Diagnostics (Shirley, NY, USA). Poly(ethylene glycol) (PEG, MW 2000), l-aspartic acid-β-benzyl ester, anhydrous 1,4-dioxane, triethylamine (TEA), hexylamine, 1-(3-aminopropyl)-imidazole, n-hexane, methanol, N,N-dimethylformamide, cellite, palladium on charcoal, N-hydroxysuccinimide (NHS), N,N’-dicyclohexylcarbodiimide (DCC), and dimethyl sulfoxide-d6 (DMSO-d6) with tetramethylsilane were purchased from Sigma-Aldrich (St. Louis, MO, USA). Triphosgene was purchased from Alfa Aesar^®^. Ethanol (EtOH), dimethylsulfoxide, and dichloromethane were purchased from Honeywell Burdick & Jackson (Muskegon, MI, USA). Cy 5.5-NHS ester was purchased from Lumiprobe (Cockeysville, MD, USA). LysoTracker^®^ Green DND-26 was purchased from Invitrogen^TM^ (Life Technologies, Thermo Fisher Scientific Inc., Waltham, MA, USA). Doxorubicin hydrochloride (DOX HCl) was purchased from Boryung Co. (Seoul, South Korea). PAIM-PEG (MW 8.5kDa, PAIM(6.5 kDa)–PEG(2 kDa)) was prepared as previously reported [43]. The conjugation ratio of the imidazole moiety to the backbone of poly(aspartic acid)-block-PEG was 60%. All other chemicals utilized were of analytical grade. KB cells were obtained from the Korean Cell Line Bank (Jongno-gu, Seoul, South Korea). RPMI 1640 medium, DPBS, penicillin–streptomycin solution, trypsin–EDTA solution, and fetal bovine serum (FBS) were purchased from Welgene (Gyeongsan-si, Gyeongsangbuk-do, South Korea). A Cell Counting Kit-8 (CCK-8) was purchased from Dojindo Molecular Technologies, Inc. (Rockville, MD, USA).

### 2.2. Synthesis of PAIM-PEG, a pH-Sensitive Polymer

PAIM-PEG (MW 6.5–2 KDa) was synthesized as previously described (Appendix A) [43]. Briefly, polymers were synthesized by ring opening polymerization with α-amino acid-N-carboxyanhydride (NCA) and prepared through sequential PEG conjugation by NHS/DCC coupling, deprotection of benzyl groups using Pd/C catalysis and hydrogen gas, and imidazole conjugation by NHS/DCC coupling. The structural properties of synthesized polymers were analyzed by ^1^H-NMR (600 MHz, DMSO-*d*_6_).

### 2.3. Preparation of Nanocluster (NC) Systems

Before loading doxorubicin into the nanoclusters, doxorubicin∙HCl (DOX∙HCl) was stirred with a two mole ratio of TEA in DMSO overnight to obtain the DOX base [52,53]. To prepare NC systems, 100, 50, and 25 μg of AuNR were mixed with 50 μg of DOX, respectively (Table 1). PAIM-PEG in DMSO was added to the mixture with 4-fold amounts of DOX. After mixing for 3 h, 7 mL of the final mixture of DMSO and distilled water (DMSO:DW = 9:1) were dialyzed (Standard Regenerated Cellulose (RC) Dialysis Tubing, MWCO 3.5 kDa, Repligen, Waltham, MA, USA) against Na_2_B_4_O_7_ buffer solution adjusted by HCl or NaOH (pH 8.2, 1 mM) for 24 h. The NC solution was filtered through a 0.45 µm filter membrane to disregard the unloaded drug and to sterilize. The NC solution was prepared directly before use.

### 2.4. Measurement of DOX and Gold Concentration

To check the actual amounts in the NC systems, AuNR was precipitated by 5000 rpm for 15 min after dilution using DMSO. The amount of loaded DOX in NC systems was determined by measuring the UV absorbance at 480 nm of the supernatants using a GENESYS 10 UV UV/Vis spectrophotometer (Thermo Scientific, Waltham, MA, USA). The concentration of AuNRs in the NC solution was measured using inductively coupled plasma mass spectrometry (ICP-MS, ICAP Q, Thermo Scientific, Waltham, MA, USA).

Drug loading content and efficiency were calculated by the following equations:Drug loading content (%) = (Weight of loaded drug in NC systems)/(Weight of total drug-loaded NC systems) × 100
Drug loading efficiency (%) = (Weight of loaded drug in NC systems)/(Weight of drug initially added to NC systems) × 100

### 2.5. Absorption Spectra

The SPR spectra of AuNR and NC systems were analyzed using a UV–Vis spectrophotometer at the wavelength range of 200–900 nm. The concentrations of DOX were 50 μg/mL.

### 2.6. Particle Size and Zeta Measurement by Dynamic Light Scattering (DLS)

The effective hydrodynamic diameter (*D*_eff._) and zeta potentials of NC Systems were measured by photon correlation spectroscopy using a Zetasizer Nano-ZS (Malvern Instruments, UK) equipped with the Multi-Angle Sizing option (BI-MAS). Software provided by the manufacturer was used to calculate D_eff._ and zeta potential values based on three measurements of each sample (*n* = 3).

NC systems at different pH conditions (0.01 wt % of AuNR) were prepared by diluting the stock 10-fold with phosphate-buffered saline (PBS, pH 7.4 and 6.5). Before size and zeta measurements were conducted, the NC solutions were incubated at room temperature for 3 h (*n* = 3).

### 2.7. Morphology of NC Systems

The morphologies of AuNR and NC systems were investigated by transmission electron microscopy (TEM) studies on a TECNAI G2 F30 S-TWIN (FEI Company, Oregon, OR, USA). A thin film of the samples stained by PTA (2%) was positioned on a copper grid covered with nitrocellulose. Before loading onto the microscope, the grid was dried in a vacuum dryer and PTA staining was inspected [54].

### 2.8. DOX Release Profile of NC Systems

For drug release assessment, NC 2 containing 40 μg of DOX was dispersed in 1 mL of PBS at different pH conditions (pH 7.4 and 6.5). The diluted solution was transferred into a Spectra/Por dialysis membrane tube with molecular weight cut-off of 6000–8000. Each membrane tube was immersed in a vial with 10 mL of PBS solution adjusted to a different pH (pH 7.4 and 6.5). The release of DOX from the NC systems was implemented under mechanical shaking (100 rpm) at 37 °C. At predetermined time intervals, the outer phase of the membrane was collected for DOX concentration analysis and substituted with the same amount of fresh medium for a sink condition (Appendix A).

### 2.9. Photothermal Effect

The photothermal effects of NC 2 were investigated under laser irradiation as a function of time, using a near-infrared (NIR) laser at an irradiation wavelength of 808 nm (Fiber-coupled laser system FC-W-808-50W 170687, UNIOTECH, Daedeok-gu, Daejeon, South Korea) and an infrared thermal imaging camera (FLIR T430sc, FLIR Systems, Inc., Wilsonville, OR, USA). Additionally, NIR thermographic images were obtained for 5 min at predetermined time intervals. One milliliter (1 mL) of PBS, DOX, AuNR, and NC 2 in PBS (equivalent amount of gold: 1 μg/mL, pH 7.4 and 6.5) were irradiated separately by an NIR laser at 808 nm (2 W/cm^2^) for 5 min.

### 2.10. In Vitro Anti-Cancer Effect

KB cells were maintained using RPMI 1640 medium supplemented with 10% fetal bovine serum and grown in a humidified incubator at 37 °C in a 5% CO_2_ atmosphere. KB cells (5 × 10^6^ cells/mL) harvested from monolayers were seeded into 96-well plates in 50 mL of RPMI 1640 for 24 h prior to cytotoxicity tests. PAIM-PEG, DOX·HCl, AuNR, and NC 2 were prepared in serum-free RPMI 1640 medium. After 24 h of serum starvation, the medium was removed from the 96-well plates for samples with different DOX concentrations (0.1 and 1 μg/mL) and the cells were incubated for 24 h.

Cell viability was measured by a CCK assay. Fresh medium containing 10 vol % of CCK solution was added to each well. The plate was incubated for an extra 3 h. The absorbance of each well was then examined on a Flexstation 3 microplate reader (Molecular Devices, Sunnyvale, CA, USA) at a wavelength of 450 nm. The viability of KB cells in each group was compared with that of non-treated cells in the same medium.

### 2.11. Cellular Uptake

The effect of nanoclusters on cellular uptake under different pH conditions was compared via flow cytometry and confocal microscopy KB cells were maintained in RPMI 1640 medium supplemented with 10% fetal bovine serum in a humidified incubator at 37 °C in 5% CO_2_. KB cells were seeded into 6-well plates (2 × 10^5^ cells/well in 3 mL media) and incubated at 37 °C in 5% CO_2_ for 24 h. The media were then removed and DOX solutions or NC 2 solutions at a concentration of 0.5 μg/mL at pH 7.4 and 6.5 were added. DOX·HCl and NC 2 in serum-free RPMI 1640 medium at pH 7.4 and 6.5 were prepared immediately before use. The pH of serum-free RPMI 1640 medium with 10 *v/v*% of PBS was adjusted with 0.1 N NaOH or 0.1 N HCl. The plate was incubated at 37 °C in 5% CO_2_ for 3 h. Cells were then washed with cold PBS three times and collected using scrapers. DOX uptake from DOX·HCl or NC 2 was subsequently determined using a BD FACS Calibur flow cytometer with Cell Quest Pro software (BD Biosciences, San Diego, CA, USA).

The pH effect on cellular uptake of DOX into KB cells was also confirmed using confocal microscopy. KB cells were grown in 6-well plates (2 × 10^5^ cells per well) containing a cover glass in each well. After 24 h, media were replaced with NC at a DOX concentration of 0.5 μg/mL in the media at pH 7.4 and 6.5. Cells fed with fresh media were designed as controls. After three hours of incubation, the media were discarded and the cells on cover glasses were washed three times with cold DPBS. They were then fixed using paraformaldehyde solution (4%) for 15 min, and subsequently washed again three times with cold DPBS. Next, cell nuclei were stained with DAPI (Thermo Fisher Scientific, Waltham, MA, USA) following the manufacturer’s protocol. Finally, cover glasses were mounted on glass slides using a Permount^®^ mounting medium (Thermo Fisher Scientific, Waltham, MA, USA). DOX uptake was observed using an LSM800 Confocal Laser Scanning Microscope (Carl Zeiss, Oberkochen, Germany).

### 2.12. Animal Care

All animal care and experiments were conducted in accordance with the National Institute of Health guidelines ‘Principles of Laboratory Animal Care’ and the ‘Animal Protection Act of the Republic of Korea’ and were approved by the Institutional Animal Care and Use Committee (IACUC) of Chung-Ang University, Seoul, Republic of Korea. Tumor xenografts were established by subcutaneously injecting 5 × 10^6^ KB cells suspended in 0.1 mL of DPBS into the right flanks of BALB/c nude mice (Orient Bio Inc., Seoul, South Korea). Tumor volume was calculated using the following equation [53,55,56,57]:tumor volume = length × (width)^2^/2.

Studies of the biodistribution and anti-cancer effects were initiated when the tumor volume reached approximately 100 mm^3^.

### 2.13. Biodistribution of NC Systems

For near-infrared fluorescence real-time tumor imaging, nanoclusters containing 10 wt % of Cy5.5-labelled NC, in 0.9 M NaCl, were injected into the tail veins of mice bearing KB tumors. The biodistribution of NC as a function of time after injection was monitored using a Fluorescence In Vivo Imaging System (FOBI system, Neo Science, Suwon, South Korea) with a red channel for Cy5.5. At 48 h post injection, the tumor and other organs were harvested to assess NC accumulation. In vivo and ex vivo fluorescence levels were analyzed with NEO image software (Neo Science, Suwon, Korea).

### 2.14. In Vivo Study: Anti-Cancer Efficacy, Toxicity, and Histological Analysis

BALB/c nude mice bearing tumors were randomly divided into nine groups including Saline without laser (Saline), Saline with laser (Saline (L+)), DOX without laser (DOX), DOX with laser (DOX (L+)), AuNR without laser (AuNR), AuNR with laser (AuNR (L+)), NC 2 without laser (NC 2), and NC 2 with laser (NC 2 (L+)). Each sample was injected intravenously into tumor-bearing mice through tail veins at a dose of 1 mg/kg DOX or 1 mg/kg AuNR. For the control group, saline (0.2 mL) was injected intravenously into the tail vein. Relative tumor volume (mm^3^) was calculated as the relative volume of a tumor at predetermined time intervals (0–28 days) to the initial tumor volume as 100. Changes in tumor size and body weight were checked every 4 days for 28 days.

Mice were sacrificed on day 28, and tumor tissue was excised, fixed in 10% formalin, processed into paraffin, and sectioned at 5 μm. Sections were stained with hematoxylin and eosin (H&E) and terminal deoxynucleotidyl transferase-mediated dUTP nick end-labeling (TUNEL) kits. All samples were examined under a Moticam Pro 205A camera coupled to a computer equipped with the software Motic Images Plus 2.0 (Richmond, BC, Canada).

## 3. Results and Discussion

### 3.1. Preparation and Optimization of NC Systems

NC systems were prepared using PAIM-PEG (pK_a_ = 6.5 [43]) and AuNR for effective combination therapy using photothermal therapy (PTT) and chemotherapy. The NC systems prepared by adding DOX and PAIM-PEG with a fixed ratio (1:4; [48]) to AuNR (100, 50, and 12.5 μg/mL) at pH 8.2 are characterized in Table 1. As the ratio of DOX and PAIM-PEG increased, the DOX loading efficiency slightly increased by 50 μg/mL of DOX. From these results, the loading capacity of NC systems for DOX was assumed to be approximately 24.5 wt %. The high DOX loading capacity in the nanoclusters may be due to hydrogen bonds and stacking interactions between the anthracyclines of DOX and imidazole groups in PAIM-PEG [58].

NC systems prepared with 50 μg/mL of DOX loading exhibited a stable nano-assembly with a thin layer of PAIM-PEG in PBS at pH 7.4, as shown in TEM images (Figure 1a). The size of nanoclusters composed of DOX, AuNR, and PAIM-PEG was found to be 15.2 ± 0.3 (nm) × 57.6 ± 5.5 (nm), 15.4 ± 2.2 (nm) × 54.4 ± 4.8 (nm), and 24.6 ± 1.6 (nm) × 65.7 ± 3.5 (nm) in AuNR, NC 1, and NC 2, respectively (width (nm) × length (nm)). As the AuNR ratio decreased, NC showed an increase in width by the increased polymeric layer of AuNR. The significant aggregation of NC 3 at pH 7.4 could be due to the destabilization of electrostatic interaction among AuNR, DOX, and PAIM-PEG. However, at an acidic pH of 6.5, the NC systems were disintegrated and formed aggregated structures due to destabilization of their hydrophobic cores by protonation of the imidazole rings and carboxyl groups in PAIM-PEG [43].

The pH sensitivity of the NC systems was also confirmed by zeta potential measurements (Figure 1b). The zeta potentials of AuNR, NC 1, NC 2, and NC 3 at pH 7.4 were found to be approximately −0.8, −5.0, −4.7, and −5.6 mV, respectively. As the pH decreased further to 6.5, the zeta potentials of the AuNR and NC systems increased to approximately +1.9, +1.3, +1.9, and +1.6 mV, respectively. This conversion of zeta potential in the NC systems is similar to that seen in polymeric micelles, likely due to the structural resemblance of the used polymers [59,60,61]. Compared with the AuNR, the NC systems had a lower zeta potential at pH 7.4, which might be due to the presence of PAIM-PEG on the AuNR surface.

### 3.2. NC Characterization

To confirm the photothermal effect, SPR peaks of the AuNR and NC systems were compared (Figure 2a). AuNR showed a SPR peak at 808 nm. NC systems also exhibited a similar SPR peak at 808 nm, while the absorbance in the range from 400 nm to 600 nm was slightly shifted. Since the peak intensity increased as the amount of PAIM-PEG increased, the slight red shift would be due to complexation by the DOX and PAIM-PEG.

Considering overall properties including loading profile, morphology, zeta potential, and SPR peak, NC 2 was selected for subsequent study including characterization, in vitro, and in vivo studies. Even though NC 1, 2, and 3 showed clear pH sensitivity by the measurement of zeta potential (Figure 1b), NC 2 showed no drawbacks compared to the others. Specifically, NC 1 showed a low loading efficiency (Table 1) and NC 3 exhibited aggregation at pH 7.4, indicating an inappropriate state for in vivo application (Figure 1a). NC 2, AuNR, and DOX in PBS (equivalent at 1 μg/mL of AuNR or DOX, the G/D/P ratio of NC 2 = 1:1:4) were exposed to an 808 nm laser at a power density of 2.0 W/cm^2^ for 5 min, respectively. Temperature variations of the solution were recorded using an infrared camera (Figure 2b). The temperature of the hybrid vesicle solution was rapidly increased to approximately 44 °C, similar to pure AuNR, which reached approximately 46 °C under the same conditions. This can be ascribed to interaction between the loaded DOX and PAIM-PEG, and the plasmonic shell might not affect the light absorption of AuNR when irradiated with an NIR laser. PBS alone showed few heating effects. The change in temperature by the 808 nm laser suggested that NC 2 could effectively generate heat for a photothermal effect to inhibit tumor growth.

To investigate the drug release profile, NC 2 was exposed to different pH conditions (pH 7.4 and 6.5) (Figure 2c). At pH 7.4, the DOX release profiles from NC systems similarly indicated less than ca. 40% maximum cumulative release over 24 h, whereas pH 6.5 led to a drastic enhancement of DOX release from NC 2. This might be attributed to altered interaction between imidazolium and carboxylate in the P(Asp-*g*-Im) block to form the pH-dependent NC core [43,49].

At pH levels above 7.4, NC 2 formed a stable nano-assembly with DOX, hydrophobic P (Asp-*g*-Im) block cores, and hydrophilic PEG coronas. NC 2 could also incorporate hydrophobic DOX in its cores by stacking and hydrophobic interactions, resulting in less release of DOX from the tight cores. Protonation and destabilization of PAIM-PEG began at pH 7.0, and NC 2 could release DOX from its destabilized cores [43]. As the pH further decreased to 6.5, alterations in nanocluster cores by predominant protonation of imidazole and carboxyl groups led to disaggregation of the assembly structures and to a decrease in the DOX carrying capacity of the NC 2, resulting in enhanced DOX release. These results suggest that the limited release of DOX from NC 2 under physiologic pH conditions (pH ≥ 7.4) would diminish the drug’s effect in blood circulation before arriving at the tumor site. The pH_ex_ in the tumor could enhance DOX release from NC systems, achieving more effective antitumor activity.

### 3.3. In Vitro Study for Cytotoxicity and Cellular Uptake

To evaluate the feasibility of NC 2 for photothermal therapy, KB cells were studied considering the cancer located on the surface of the human body that can be treated by PTT [62,63]. The cell viabilities of NC 2 were evaluated for the synergistic effect of DOX and AuNR. AuNR showed toxicity by a photothermal effect induced by the 808 nm laser (Figure 3a), whereas DOX exhibited similar cytotoxicity regardless of the light radiation. NC 2 exerted synergistic cytotoxicity at 0.1 μg/mL of AuNR and DOX even without laser irradiation (combination index (C.I.) value at ED_50_: 0.082 < 1; C.I. was analyzed by Compusyn software.) PAIM-PEG was not toxic with the laser due to the controlled DOX release by the pH-sensitive polymer. NC 2 also showed clearly enhanced cytotoxicity.

To evaluate the synergistic effect, cellular uptake of DOX under different pH conditions was measured in live KB cells by flow cytometry. Doxorubicin uptake was analyzed by flow cytometry confocal laser microscopy (Figure 3b,c). When KB cells were treated with NC 2 at pH 6.5, 1.24-fold fluorescence intensity was observed compared with at pH 7.4 (Figure 3b). DOX in NC 2 could be uptaken 1.85-fold more than in DOX alone. Consistent with flow cytometry, confocal imaging of KB cells indicated successful cellular uptake of doxorubicin. Visualizing endosomal disruption by the pH-sensitive nanocarrier, the endosomal compartment was observed by confocal microscopy using different fluorescent dyes (doxorubicin and lysotracker Green DND-26). LysoTracker^®^ Green DND-26 showed green fluorescence located in the cellular endosomal and lysosomal compartments (Figure 3c). The distribution of LysoTracker in cells indicated the disruption of endo-lysosomal compartments by the ‘proton sponge effect’ [64]. Moreover, DOX fluorescence did not significantly overlap endolysosomal compartments. The DOX loaded in pH-sensitive NC 2 could be released into the cytoplasm through breach of endolysosomal compartments by protonation of PAIM-PEG [65,66]. Therefore, the high cytotoxicity of NC 2 would be due to high intracellular distribution by pH-dependent release.

### 3.4. In Vivo Evaluation of NC

The tumor targeting of NC 2 was studied in tumor-bearing nude mice with high-resolution fluorescent imaging using Cy 5.5 labelled on the PAIM-PEG layer of NC 2 [49]. The results showed that NC 2 gradually accumulated at tumor sites by 48 h (Figure 4a). At 24 h, a marked concentration at tumor sites was evident. NC 2 remained accumulated at tumor sites for an additional 24 h. The biodistribution of NC 2 was examined with excised organs from the nude mice sacrificed after 48 h. Representative fluorescence images of various organs indicated that most of the NC 2 had accumulated at tumor sites and kidneys (Figure 4b,c).

Tumor growth inhibition was studied using a tumor-bearing mouse model with the same cell line (Figure 5). For the in vivo study, DOX, AuNR, and NC 2 were utilized with equivalent to 1 mg/kg of DOX or AuNR. Compared with DOX or AuNR with or without light, NC 2 with 808 nm laser irradiation obviously showed superior tumor growth inhibition (Appendix A). With the laser ablation by PTT, NC 2 could overcome the barrier and penetrate into tumor tissue with a three-dimensional (3D) structure [67,68,69]. It would enable significant tumor growth inhibition by pH-dependent drug release at the intracellular compartment. Corresponding to the in vitro study results, NC 2 with light irradiation demonstrated more effective tumor growth inhibition with temporal tumor eradication (Days: 8–16, Figure 5a), though tumors relapsed in groups treated by NC 2 with laser irradiation. Even though NC 2 was highly accumulated in the kidney, changes in the body weight of nude mice treated with DOX, AuNR, and NC 2 were negligible, indicating that NC 2 showed no obvious toxicity (Figure 5b). Previous reports showed biological safety, despite the accumulated AuNR in tissues such as kidney and liver [70]. In addition, this lower amount of DOX and AuNR for synergistic effects could decrease the systemic toxicity. As shown in Figure 5c, the histologic studies of tumor tissue in mice treated with NC 2 and laser irradiation (NC 2 (L+)) showed apoptosis and necrosis, with much more significant destruction than the NC 2 or AuNR (L+) groups (Red arrows). No obvious pathological abnormalities or lesions were seen in the Saline, Saline (L+), DOX, DOX (L+), and AuNR without laser groups. These results are consistent with the in vitro results, indicating that NC 2 may inhibit tumor growth in vivo and in vitro.

## 4. Conclusions

In conclusion, an NC system could minimize systemic toxicity at a physiologic pH by a robust nano-assembly. The NC system could also improve antitumor efficacy by increased accumulation and release of DOX at pH_ex_ and pH_en_ with locally applied near-infrared light. The NC system combining DOX, gold nanorods, PAIM-PEG, and light irradiation would be an on-demand strategy for treating tumors using combination therapy with less systemic toxicity and an improved therapeutic effect. To perfectly achieve the sterilization of the formulation, further studies would be needed.

## Figures and Tables

**Figure 1 pharmaceutics-12-00839-f001:**
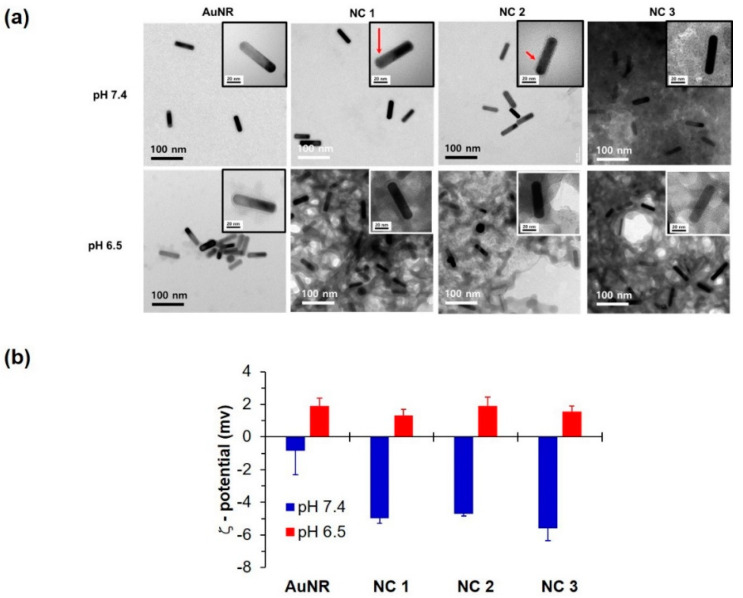
Characterization of nanocluster (NC) systems: (**a**) morphologies by transmission electron microscopy (TEM) at pH 7.4 and 6.5, and (**b**) zeta potential at pH 7.4 and 6.5 of gold nanorod (AuNR) and NC systems at different pH conditions (pH 7.4 and 6.5).

**Figure 2 pharmaceutics-12-00839-f002:**
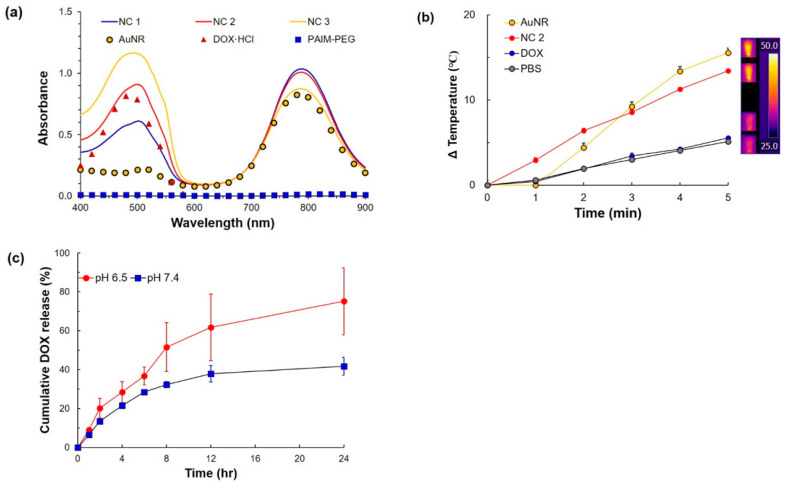
Photothermal effect: (**a**) absorption spectra, (**b**) heat generation by NIR laser (AuNR 1 μg/mL, 808 nm, 2.0 W/cm^2^), and (**c**) DOX release at different pH conditions (pH 7.4 and 6.5).

**Figure 3 pharmaceutics-12-00839-f003:**
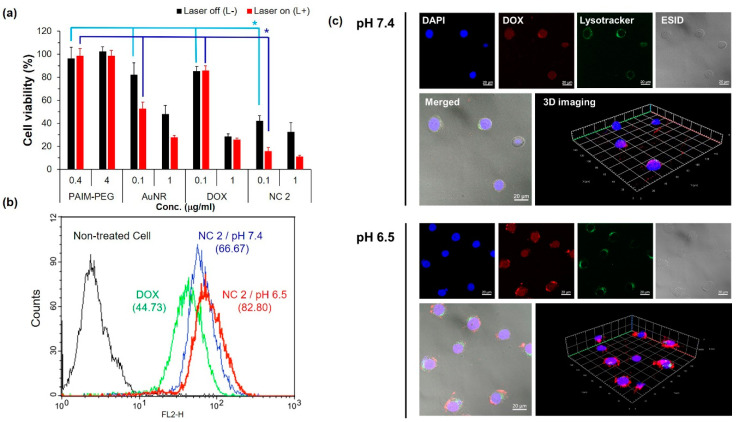
In vitro study of NC: (**a**) Cell viability of KB cells treated with poly(aspartic acid-graft-imidazole)-PEG (PAIM-PEG), AuNR, DOX, and NC 2 with or without an NIR laser (808 nm, 2 W/cm^2^, 2 min); * indicates a combination index (C.I.) < 1.0. Nanocluster effect on cellular uptake of doxorubicin determined by (**b**) flow cytometry and (**c**) confocal microscope.

**Figure 4 pharmaceutics-12-00839-f004:**
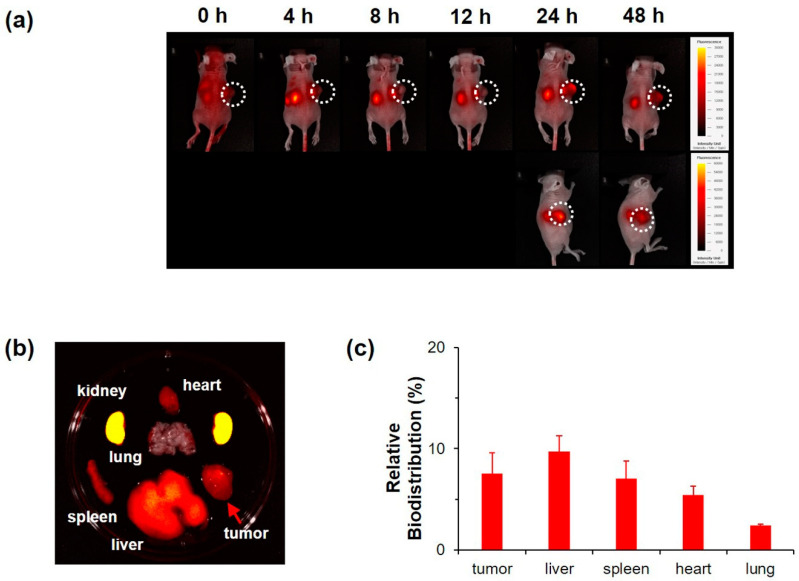
Non-invasive in vivo fluorescent imaging of Cy5.5-tagged NC 2 after intravenous injection into tail veins of single KB tumor-bearing nude mice. (**a**) Whole body imaging at predetermined time points after i.v. injection. (**b**) Ex vivo optical and fluorescent imaging of tumor and organs obtained 24 h post injection. (**c**) Relative biodistribution of NC 2 by quantitative fluorescence intensity (FI) of tumors and main organs.

**Figure 5 pharmaceutics-12-00839-f005:**
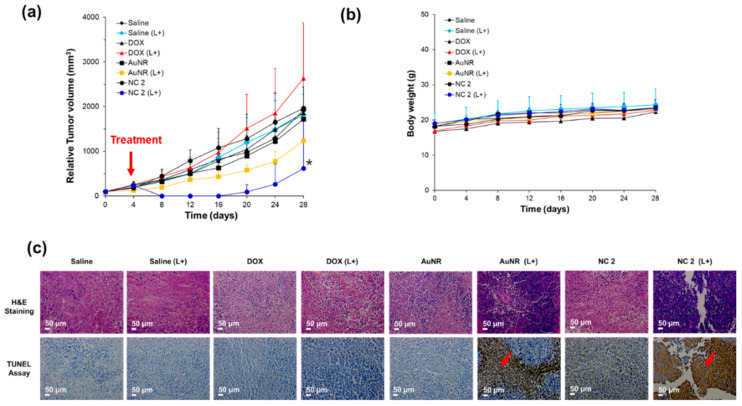
In vivo photothermal effect of NC 2: (**a**) tumor efficacy, (**b**) tumor toxicity, and (**c**) histologic study using Haemotoxylin and Eosin (H&E) staining and a Tunnel assay of tumor tissues on Day 28 (* indicates tumor relapse, *n* = 1). Red arrow indicates the treatment point of therapeutic assay to all experimental groups.

**Table 1 pharmaceutics-12-00839-t001:** Drug loading profile of nanoclusters.

Materials	AuNR			DOX			* G/D/P Ratio
Code	Target Content (wt %)	Loading Content (wt %)	Loading Efficiency (wt %)	Target Amount (wt %)	Loading Content (wt %)	Loading Efficiency (wt %)	
**NC 1**	50	32.9	65.8	25	23.9	95.5	2:1:4
**NC 2**	25	22.6	90.3	25	24.5	98.1	1:1:4
**NC 3**	12.5	11.2	89.5	25	22.7	90.6	0.5:1:4

* G/D/P Ratio indicates the ratio between gold nanorods (AuNR) (G): doxorubicin (DOX) (D): poly(aspartic acid-graft-imidazole)-PEG (PAIM-PEG) (P).

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
