# Peer review of "An On-Demand pH-Sensitive Nanocluster for Cancer Treatment by Combining Photothermal Therapy and Chemotherapy"

_pharmaceutics, 2020, doi:10.3390/pharmaceutics12090839_

Round 1

Reviewer 1 Report

In the paper,author structure a pH-sensitive nanocluster and present effect of nanocluster on combined treatment of cancer. The work is interesting to the design of drug delivery systems of NPs. Beside this I have recommendation to revise the paper.

1. The nanocluster was accumulated in kidney quite obviously, nevertheless, in figure 5,the result of kidney pathological examination are not presented, please added.

2. Safety risk of the obvious accumulation in kidney of the nanocluster by a single injection is necessary to discuss.

Author Response

  1. The nanocluster was accumulated in kidney quite obviously, nevertheless, in figure 5, the result of kidney pathological examination is not presented, please added. Safety risk of the obvious accumulation in kidney of the nanocluster by a single injection is necessary to discuss.

â–¶ Thank you for your valuable comment. Based on the previous report, gold nanorod induce few toxicities with preferential accumulation in kidney (Bailly, Anne-Laure, et al. "In vivo evaluation of safety, biodistribution and pharmacokinetics of laser-synthesized gold nanoparticles." Scientific reports 9.1 (2019): 1-12.). As shown on the subset of Figure 4c, nanoclusters with gold nanorod was highly accumulated on kidney. Even though high accumulation, there was no significant toxicity in vivo (Figure 5b). Considering the accumulation on kidney, the manuscript was revised as shown below:

page 10, line 340-344

Even though NC 2 was highly accumulated in kidney, changes in the body weight of nude mice treated with DOX, AuNR, and NC 2 were negligible, indicating that NC 2 showed no obvious toxicity (Figure 5b). Previous reports showed the biological safety even though the accumulated AuNR in tissues such as kidney and liver [72]. In addition, this less amount of DOX and AuNR for synergistic effects could decrease the systemic toxicity.

Reviewer 2 Report

The article introduced a novel pH-sensitive polymer PAIM-PEG-modified nanorod and loaded with Doxorubicin base to achieve a combinatorial anticancer effect from both nanorod and doxorubicin moieties. The studies are well-designed and there are sufficient results to support the authors' conclusion. The manuscript is overall well-written. However, there are some points that need to be elucidated before acceptance of the paper. 

1) The idea of combining pH-sensitivity, Dox and nanorods together has been reported in 2018 by Jin Chen in Bioactive Materials, volume 3, issue 3. The difference is that Chen's group used a pH-sensitive linker between Dox and nanorod, while here a pH-sensitive polymer mesh is coated on the surface of nanorod. What does the author think about comparing these two models? Is adding a layer of pH-sensitive polymer an advantage over pH-sensitive linker, in terms of drug loading, release profile, biocompatibility? 

2) Since the Dox base was loaded in the matrix of polymer and the combinatorial effect comes from Dox and nanorod, why not delivering DOX-loaded PAIM-PEG micelles and gold nanorods separately? The author is expected to justify in introduction or discussion. 

3) The PAIM-PEG polymer has been shown to have a very good pH sensitivity. The author also introduced two pH condition, tumor extracellular and endosomal. So the question is where the release of Dox do the authors expect to happen, extracellularly or intracellularly? If the release is extracellular, what's the author's opinion on tumor penetration in a 3D structure? 

4) The method parts require further clarifications. On line 86, the M.W. 6.5-2 KDa is confusing. Is it 6.5 KDa- 2KDa or 6.5 Da to 2KDa. How monodisperse is the MW of the polymer?

5) On line 95, what's the material of dialysis membrane? The author should reveal that as some materials are not compatible with DMSO:DW=9:1.

6) On line 96 and 166, the author indicated HCl or NaOH. Please explain why HCl or NaOH was used to reach pH 8.2.

7) On line 97, the author indicated a "micelle solution". According to my understanding, the solution contains gold nanorod, Dox and PAIM-PEG. If the PAIM-PEG can coat on the gold nanorod, it's inappropriate to refer the solution as "micelle solution".

8) On line 250, the author should explain more quantitatively on the selection among NC1, NC2 and NC3, because to some aspects they are very close to each others. 

9) In section 2.3, the gold nanorod and dox amounts used do not match with what illustrated in Table 1, which matched with the result section. The authors need to consistently report the numbers. It will also be a good idea to refer the "%" as "wt%" in table 1. 

10) In section 2.8, the PBS solution with different pH at a dilution factor of 10 was used as sink conditions. The author needs to show what's the saturated concentration of Dox base under this solution because the Dox base is hydrophobic and may not be able to get dissolved in PBS only. Otherwise it couldn't be used as a sink condition. 

11) In section 2.12, what's the tissue origin of the KB cells? Is it the Hela contaminants? What's the rationale behind picking up this cell line? 

12) In section 3.1, the first paragraph overlapped with the method section. 

13) The author should provide a higher resolution figure of Figure 3c or make it a separate figure since a lot of conclusions, such as endosomal escape, reply on the high resolution of this figure. 

14) Is the y-axis unit mis-labeled in Figure 5a? 1000% increase would be already 1000 mm3. 

15) The legends of Figure 4 and Figure 5 are wrong.

16) The method section has demonstrated that the tumors are inoculated on the right flank of mice. In Figure 4 a, what are the red spots on the left flanks of mice? Are they tumors? Why are they on the left flanks?  

17) Statistics explanation is missing throughout the paper. Also, please include statistics analysis in necessary places such as Figure 3. 

I would expect the author's thoughts and more discussion on the first 3 bullet points. All other points are necessary to be corrected or explained before publication.

Author Response

I have attached the response as below.

Reviewer 3 Report

The manuscript entitled “On-demand pH-sensitive nanocluster for cancer 3 treatment by combining photothermal therapy and 4 chemotherapy” with minor revisions.

  1. The authors have stated Photothermal therapy (PTT) causes cytotoxicity against malignant and diseased cells by combining a photosensitizer and a light source. How the present study deals with this aspect of cytotoxicity caused by PTT.
  2. Improvement in the font size of Scheme. (a)
  3. Some more details of Synthesis of PAIM-PEG, a pH sensitive polymer should be provided though reference has been cited.
  4. Elaborate in detail the synthesis. Preparation of Nanocluster (NC) Systems, Mixing time should be mentioned.

  1. How was the formulation sterilized? What was the storage condition?

  1. The TEM shows the particles formed in rod shaped, What is the exact size obtained by TEM. The TEM Figure is not discussed in Results. Why nanocluster name was selected for these rod shape nanoparticles? The authors should give very clear explanation.

  1. Confocal images can be improved.

Author Response

I have attached the responses as below.
